



# Topologically Optimized Magnetic Lens for MR Applications

Sagar Wadhwa[1], Mazin Jouda[1], Yongbo Deng[2], Omar Nassar[1], Dario Mager[1], and Jan G. Korvink[1]

[1]Institute of Microstructure Technology, Karlsruhe Institute of Technology, Hermann-von-Helmholtz-Platz 1, 76344 Eggenstein-Leopoldshafen, Germany
[2] State Key Laboratory of Applied Optics (SKLAO), Changchun Institute of Optics, Fine Mechanics and Physics (CIOMP), Chinese Academy of Sciences, Dongnanhu Road 3888, Changchun 130033, China

**Correspondence:** Jan G. Korvink (jan.korvink@kit.edu)

**Abstract.** Improvements to the signal-to-noise ratio of magnetic resonance detection leads to a strong reduction in measurement time, yet as a sole optimization goal for resonator design it would be an oversimplification of the problem at hand. Multiple constraints, for example for field homogeneity, and sample shape, suggests the use of numerical optimization to obtain resonator designs that delivers the intended improvement. Here we consider the 2D Lenz lens as a sufficiently broad-band flux transforming interposer between the sample and an RF circuit, as a flexible and an easily manufacturable device family with which to mediate different design requirements. We report on a method to apply topology optimization to determine the optimal layout of a Lenz lens, and demonstrate realisations for both low (45 MHz), and high frequency (500 MHz) NMR.

## 1 Introduction

### 1.1 Signal to Noise Ratio *(SNR)* in Magnetic Resonance (*MR*)

Nuclear *MR* spectroscopy and imaging are powerful tools for determining the molecular structure of chemical substances, or for studying the anatomy of organisms. For *MR* measurements, it is important to achieve a high *SNR* to obtain a high-resolution spectrum or a highly resolved image, and also leads to a reduction in the overall measurement time. The relationship between the *SNR* and the magnetic field produced by the coil was derived by Hoult et al. (1976) and can be reduced to

$$SNR \propto \frac{V B_1}{I \sqrt{R}}, \tag{1}$$

where $B_1$ (in T) is the radiofrequency magnetic field produced by the coil in a direction normal to the polarising field $B_0$, $V$ (in $m^3$) is the sample volume, $I$ (in A) is the current flowing through the coil, and $R$ (in ohms) is its AC resistance. From the *SNR* equation (Equation 1) it can be deduced that, for a given magnitude of $B_1$, and a constant $I$ flowing through the coil, the *SNR* depends directly on the sample volume, therefore, for a smaller volume of the sample, the *SNR* degrades significantly. The filling factor of the coil, which is the geometrical relation between the sample volume, and the size of the useful $B_1$-volume of





the coil, further limits the sample volume for a fixed size of the coil. To improve the *SNR*, it is important to improve the filling factor by increasing the magnetic field penetration through the sample, whilst maintaining constant value of the $B_1$-field.

For the filling factor—the ratio of the geometrical size of the coil with respect to the sample volume—it is self-evident that reducing the size of the coil will improve its SNR, since the strength of Farady induction falls off with distance. The minia-

turization of coils, as for example discussed by Korvink et al. (2019), comes with its additional advantages and limitations. The efficiency of the coil increases as the desired magnetic field can be achieved with lower electrical power, yet the use of relatively bulky capacitors for matching and tuning cannot be avoided, since the electrical length scales inversely with the frequency in Maxwell's equation. To tune and match the coil at the frequency of operation, the electrical connections for the capacitors needs to be established close to the coil, which is disadvantageous.

In some cases the reception coil and its capacitors cannot therefore be placed too close to the sample or specimen, for e.g. when performing *MRI* on small living organisms, or for sensitive spectroscopy of small samples. In such scenarios, the improvement of the filling factor is remedied by using a Lenz Lens (*LL*) Spengler et al. (2017) which focuses the magnetic field produced by a larger coil into a smaller sample region. Their working principle simply follows Lenz's law of induction, which augmets Faraday's law. The transmitter coil induces a current in the outer loop of the *LL*, and by design forces the

induced current to also flow to an inner loop, but directed in the opposite sense. The dimension of the inner loop is such that it encircles the sample completely. This simultaneously results in a localized magnetic field amplification within the inner loop, and a zeroing of the field in the outer loop. Since a *LL* is broad-band up to its high resonance frequency (usually in the GHz range), these devices can be used over a wide range of frequencies without additional tuning. The field amplification produced by a *LL* depends on the area ratio of the outer to the inner loops. When the *LL* is limited by the available working space, the

total magnification that can be achieved is lowered as was shown by Jouda et al. (2017).

If we focus our attention towards the *LL*, its design needs to be further investigated to improve the amplified field uniformity, and to increase the field amplification, for cases where the geometrical space for the *LL* is limited. Although it was shown by Jouda et al. (2017) that by tuning and matching the *LL* at the frequency of operation, this improved the signal acquired significantly, even for those cases where the design space is constrained, it comes at the cost of losing the broad-band nature

of the *LL* and adds the difficulty in maintaining the *Q* factor of the coil/lens arrangement due to the resonance splitting effect. Since the solution to the problem lies in finding the best possible topology of the metal structure to overcome these issues, it prompted us to formulate a method where the tailoring of the field could be controlled mathematically, while searching for the optimal design.

In this paper, we now explore the use of computational optimization to "discover", via inverse design, a novel distributed

metallic track arrangement that produces the same effect as a Lenz lens. The computational procedure will aim to maximise the magnetic field flux (i.e., the lensing effect) in the sample, and at the same time, aim for a flux distribution that is as constant as possible. As will be shown, these two requirements are in conflict, so that a supervisor will have to balance the requirements depending on the application. Furthermore, the design will depart considerably from the Lenz lens topology, and may require additional constraints to ensure manufacturability.





## 1.2 Topology Optimization

Topology optimization has been used in various fields for inverse material design, such as for acoustics (Dühring et al. (2008)), mechanical structures (Bendsøe et al. (1988)), electromagnetics (Sigmund et al. (2008)), thermodynamics (Gersborg-Hansen et al. (2006)), fluidics (Zhou et al. (2008)), and permanent magnetic system (Lee et al. (2010)) to name a few.

In the field of electromagnetics, topology optimization has been explored for applications such as photonic crystals (Sigmund et al. (2008)), dielectric clocks (Deng et al. (2018)), beam splitters (Piggott et al. (2015)), antennas (Zhou et al. (2010)), surface plasmonics (Andkjær et al. (2010)), and more. In most of these cases, the electrical permittivities and magnetic permeabilities of the material were used as a function of space to obtain a material distribution.

However, in surface plasmonics (Andkjær et al. (2010)), and antenna design (Zhou et al. (2010)), the conductivity value of the material was used to realize the desired structures. For the excitation of surface plasmons, the frequency of operation was in the THz range, for which the normal component of the electric field on any boundary is negligible, and the metal domain can be truncated by applying a perfectly electric conductor boundary condition (*PEC*). When considering metals in the radio and microwave range (3 MHz-300 GHz), it can no longer be considered as a *PEC* due to the skin depth effect, for which the current penetrates up to a certain thickness of the metal before it decays completely.

Aage et al. (2010), introduced a method to implement impedance boundary conditions (*IBC*) for the inverse material design of metals in the microwave range which takes the skin depth effect in to account. During the post-processing, they used the *PEC* condition on the metal's boundaries with which to compute solutions.

For our problem formulation we chose the conductivity function as a material property in the domain rather than on the boundary. The conductivity range between which the material property is interpolated was in the order of $10^7$, which is similar to the ratio of the conductivity of Copper (*Cu*) and that of free space. Since the free space conductivity was set close to zero rather than exactly zero, it forced some portion of the current to be normal at the boundary of free space before it decays. This imitates the behaviour of an *IBC*, which is then used to generate the material design. For post processing, the *IBC* was imposed on the boundaries of the optimized Lens geometries to measure the actual enhancement of the magnetic field.

Using the methodology presented in Section 2, 3, and 4, we optimized the magnetic lens for a Larmor frequency of *(i)* 45 MHz, and *(ii)* 500 MHz, where it was assumed that the excitation $B_1$-field is oscillating along an axis perpendicular to the *OL*. The obtained geometries were then characterised, and the simulation results were verified by *NMR* experiments described in Section 5.2. For the 45 MHz *OL* design, NMR measurements were performed on a 1.05 T preclinical *MRI* machine (Bruker, ICON). The 500 MHz *OL* measurements were performed on an 11.7 T vertical wide-bore superconducting NMR spectrometer (Bruker AVANCE III). To verify the enhancement of the magnetic field and improvement in *SNR*, a series of nutation spectra of a destilled water sample were taken, both with and without an *OL*.





## 2 Methodology


In this section we consider the electromagnetic wave equation that governs the behaviour of the device, derive the equations of the material distribution method, and formulate the objective function with constraint equations with which to obtain an optimized geometrical configuration corresponding to a spatial material distribution.

The time varying magnetic field $\boldsymbol{B}(t)$ and electric field $\boldsymbol{E}(t)$ can be defined in terms of a time varying magnetic vector $\boldsymbol{A}(t)$

and an electric scalar potential $\phi(t)$. Assuming time-harmonic behaviour proportional to $e^{j\omega t}$, where $j$ represents the imaginary unit $(\sqrt{-1})$, $\omega$ represents the angular frequency, and $t$ the time. $\boldsymbol{B}$, and $\boldsymbol{E}$ in the frequency domain are defined as

$$\boldsymbol{B} = \nabla \times \boldsymbol{A}, \tag{2}$$

$$\boldsymbol{E} = -\nabla \phi - j\omega \boldsymbol{A}. \tag{3}$$

Here $\nabla \overset{\text{def}}{=} (\partial/\partial x, \partial/\partial y, \partial/\partial z)$ represents the gradient operator in a Cartesian coordinate frame of reference. By substituting

$\boldsymbol{A}$ and $\phi$ from Equation 2, and 3 into Maxwell's equations, and simplifying the result by fixing the Lorenz gauge in order to obtain a unique solution, the modified wave equation with the divergence free condition becomes

$$\nabla^2 \mathbf{A} - \omega^2 \mu_r \mu_0 \epsilon_r \epsilon_0 \mathbf{A} = -\mu_r \mu_0 \sigma \nabla(\phi) \quad \text{in } \Omega, \tag{4}$$

$$\nabla \cdot \nabla(\phi) = 0 \quad \text{in } \Omega. \tag{5}$$

Here $\mu_r$, and $\epsilon_r$ represents the relative permeability and permittivity of the propagation medium. $\mu_0$ and $\epsilon_0$ represents the free

space permeability and permittivity. and $\sigma$ denotes the conductivity of the medium. $\Omega \subset \mathbb{R}^3$ is the entire computational domain. We will assume that the material properties are isotropic, but this is not a fundamental restriction.

Figure 1 is a schematic of the rectangular computational domain $\Omega$. The light blue region in te centrerepresents the target domain ($\Omega_T$), where the magnetic field is tailored to be maximized and evenly spread. The region in copper represents the design domain ($\Omega_D$), where the material is interpolated between $Cu$, and air to achieve the desired field amplification.

$\Omega_D$ is enclosed by an infinite element domain ($\Omega_I$), indicated in gray. $\Omega_I$ is stretched rationally by a factor of $10^3$ such that the magnetic vector potential ($\boldsymbol{A}$) decays exponentially as a function of distance from the enclosed domain $\Omega_D$. Using a method reported in Demkowicz et al. (1998), the computation of the wave propagation in $\Omega_I$ is fully specified by

$$\nabla^2 \boldsymbol{A} - \omega^2 \mu_r \mu_0 \epsilon_r \epsilon_0 \boldsymbol{A} = 0 \quad \text{in } \Omega_I, \tag{6}$$

$$\boldsymbol{n} \times (\nabla \times \boldsymbol{A}) = g \quad \text{on } \partial\Omega_D \cup \partial\Omega_I, \tag{7}$$

$$\boldsymbol{e_r} \times (\nabla \times \boldsymbol{A}) - j\omega \boldsymbol{A} = O(r^{-2}) \quad \text{as } r \to \infty, \tag{8}$$

where $\boldsymbol{n}$ is the normal outward vector on the boundary of $\Omega_T$ and $\Omega_I$, $g$ is the tangential magnetic vector potential on $\partial\Omega_D \cup \partial\Omega_I$, and $\boldsymbol{e_r}$ is the unit vector in the radial direction. Equation 8 represents the exponential decay of the magnetic field. The entire computational domain is truncated by a magnetic insulation boundary condition such that

$$\mathbf{n} \times \mathbf{A} = 0 \quad \text{on } \partial\Omega \tag{9}$$





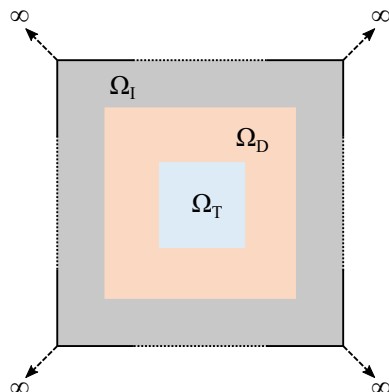

**Figure 1.** Sketch of the computational domain for the topology optimization of the magnetic lens. The domain in blue represents the target domain ($\Omega_T$) where the magnetic field was focused. The focusing of the magnetic field was achieved by the material interpolation between $Cu$, and air in the design domain ($\Omega_D$) represented in orange. $\Omega_D$ was enclosed by an infinite element domain ($\Omega_I$) in gray. The entire computational domain was $\Omega = \Omega_T \cup \Omega_D \cup \Omega_I$. $\Omega$ was truncated by imposing a Magnetic Insulation Boundary Conditions on its boundaries ($\partial\Omega$).

The topology optimization is achieved based on an adjustable, spatially varying material property. We selected the conductivity of the medium as a function of the spatial coordinates, which interpolates between free space, and $Cu$. To find the values of the conductivity in $\Omega_D$, a design variable ($\gamma$) was introduced such that $\gamma \in [0,1]$, where zero represents free space, and unity represents $Cu$. The variable $\gamma$ is filtered using a Helmholtz filter equation (Lazarov et al. (2011)). Based on the finite element mesh size used for the computation, the radius filter was set to twice the mesh size to avoid ambiguity during the decision

phase of material interpolation. The filter equations are

$$-r^2 \nabla \cdot \nabla \gamma_f + \gamma_f = \gamma \quad \text{in } \Omega_D, \tag{10}$$

$$\nabla \cdot \gamma_f = 0 \quad \text{on } \partial\Omega_D, \tag{11}$$

where $r$ is the radius filter, which sets the minimum feature size of the $Cu$ and $\gamma_f$ is the filtered design variable. As $\gamma$ can have any values between zero and one, it's important to find a solution such that it converges to either of these values. Therefore, in

order to reduce intermediate grayscale values, so as to achieve a high contrast material distribution, $\gamma_f$ was projected using a hyperbolic tangent function (Guest et al. (2004); Wang et al. (2011))

$$\gamma_p = \frac{\tanh(\beta\xi) + \tanh(\beta(\gamma_f - \xi))}{\tanh(\beta\xi) + \tanh(\beta(1-\xi))}, \tag{12}$$

where $\beta$ is the projection slope, $\xi \in [0,1]$ is the projection point, and $\gamma_p$ is the calculated projected design variable. From Figure 2(a), it can be observed that $\gamma$ can be continuously varied between a linear interpolation and a unit step function based

on the value of $\beta$. To achieve a robust algorithm, $\beta$ was incremented after a fixed number of iterations. After solving for $\gamma_p$, it is used in the conductivity function to find the density distribution of $Cu$ in $\Omega_D$, which is realized by a combination of

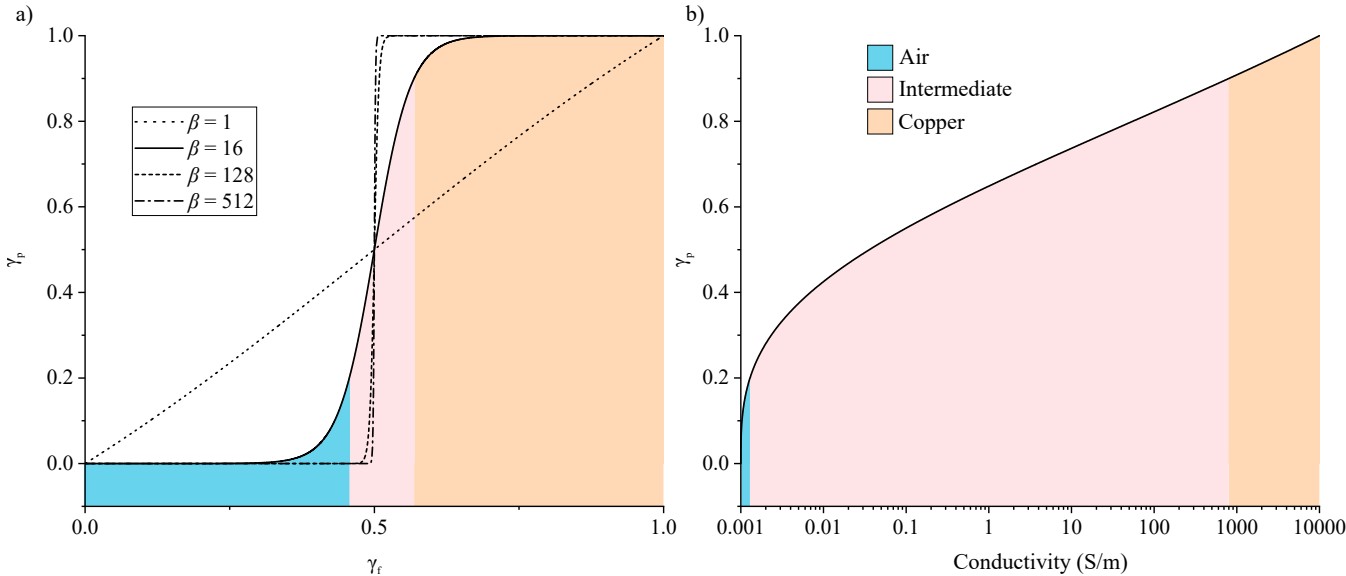

**Figure 2.** (a) Relationship between $\gamma_f$, and $\gamma_p$ for different values of $\beta$. (b) Conductivity values as a function of the projected design variable ($\gamma_p$) on a logarithmic scale calculated from Equation 13. As $\beta$ increases, the intermediate contour in (a) decreases, forcing $\gamma_p$ to converge to either air or *Cu*. In (b) the air and *Cu* contours represent the range of conductivity values it can have. With increasing $\beta$, the contour area for air and copper decreases, forcing it to have a unique value.

logarithmic, and power law expressions (Diaz et al. (2010); Deng et al. (2016)),

$$\sigma(\gamma_p) = 10^{\log \sigma_m - (\log \sigma_m - \log \sigma_{air})(1-\gamma_p^p)/(1+\gamma_p^p)} \quad \text{in } \Omega_D, \tag{13}$$

where $\sigma_m$ is the conductivity of *Cu*, $\sigma_{air}$ is the conductivity of the air, and *p* is the penalization factor. As can be seen from Figure 2(b) the conductivity varies between metallic values for $\gamma_p$ close to 1, tends to that of free space for values of $\gamma_p$ close to 0.

After defining the wave equation, and the material distribution equations, control equations were defined to meet the requirements of the *MR* experiments, *i.e.*, to have a uniform $\boldsymbol{B}_1$ distribution along with its enhancement. From the equation of the flip angle

$$\alpha = \frac{\gamma_r \boldsymbol{B}_1 \tau}{2\pi}, \tag{14}$$

where $\gamma_r$ is the gyromagnetic ratio in $\mathrm{MHzT}^{-1}$, if $\boldsymbol{B}_1$ is not uniform, the net magnetization will tip by different angles at different voxel positions, which is to be minimized. This reduces the total signal generated. The uniformity of the magnetic





field distribution (without affecting the enhancement) were controlled by two equations,

$$\int_{\Omega_T} \left( \frac{\nabla \times \boldsymbol{A}}{\nabla \times \boldsymbol{A}_{\text{ref}}} - 1 \right)^2 d\Omega_T, \quad \text{(uniformity control equation),} \tag{15}$$

$$\int_{L_\xi} \left( \frac{\nabla \times \boldsymbol{A}}{\nabla \times \boldsymbol{A}_{\text{ref}}} \right) d\zeta \geq 1 \quad \text{for } \zeta \in (x,z), \quad \text{(field amplification equation)} \tag{16}$$


where $\boldsymbol{A}_{\text{ref}}$ is the large reference magnetic vector potential calculated from Equation 2 for the reference magnetic field and $L_\xi$ is one or more diagonal lines in $\Omega_T$. Equation 15 is an error function of the computed magnetic field and the reference magnetic field, and Equation 16 is an inequality function which leads to the computed field to evolve towards the reference field, hence its enhancement. Equation 16 provides computational freedom to find the *Cu* design while satisfying its condition. At the same time Equation 15 is used as an objective function, to ensure that the *Cu* distribution minimizes it. The total area that *Cu* could occupy in $\Omega_D$ was controlled by


$$\int_{\Omega_D} (\gamma_p) d\Omega_D \leq 0.5, \tag{17}$$

which is a 50% proportion of $\Omega_D$.

From the above discussion, the optimization problem formulated can be said to be (oxymoronically) a minimization-maximization process, where the minimization of the objective function $J$ leads to a maximization of the magnetic field while maintaining its uniformity. To summarize, the goal of the optimization was to


$$\text{find } \gamma \in [0,1] \text{ to minimize } J = \int_{\Omega_T} \left( \frac{\nabla \times \mathbf{A}}{\nabla \times \mathbf{A}_{\text{ref}}} - 1 \right)^2 d\Omega_T \quad \text{(field uniformity control equation),}$$

$$\text{subject to} \begin{cases} \nabla^2 \mathbf{A} - (\omega \mu_r \mu_0 \epsilon_r \epsilon_0)^2 \mathbf{A} & = -\mu_r \mu_0 \sigma(\gamma_p) \nabla(\phi), \text{ in } \Omega, \quad \text{(wave propagation equation),} \\ \nabla \cdot \nabla(\phi) & = 0, \text{ in } \Omega, \quad \text{(divergence-free condition),} \\ -r^2 \nabla \cdot \nabla \gamma_f + \gamma_f & = \gamma, \text{ in } \Omega_D, \quad \text{(Helmholtz filter equation),} \\ \nabla \cdot \gamma_f & = 0, \text{ on } \partial \Omega_D \quad \text{(design variable divergence free condition),} \\ \int_{L_\zeta} \left( \frac{\nabla \times \mathbf{A}}{\nabla \times \mathbf{A}_{\text{ref}}} \right) d\zeta & \geq 1, \quad \zeta \in (x,z) \quad \text{(field enhancement equation)} \\ \int_{\Omega_D} (\gamma_p) d\Omega_D & \leq 0.5 \quad \text{(material control equation).} \end{cases} \tag{18}$$

The material properties applied to the different domains are represented in Table 1. The implementation of the algorithm, and the results obtained, are discussed in the next section.

## 3 Numerical Implementation


The optimization computations were carried out in the commercial software COMSOL MULTIPHYSICS (V5.4) using its AC/DC, and Optimization modules. The simulations were computed using Intel(R) Xeon(R) Silver 4210 CPU with processing speed of 2.2 GHz, and RAM size of 64 GB on a 64-bit Windows 10 operating system.



**Table 1.** Material properties assigned to the different domains for the computation of the optimized geometrical configiration

| Domain | $\mu_r$ | $\epsilon_r$ | $\sigma$ |
|--------|---------|--------------|----------|
| $\Omega_T$ | 1 | 80 | 0 |
| $\Omega_D$ | 1 | 1 | $\sigma(\gamma_p)$ |
| $\Omega_I$ | 1 | 1 | 0 |

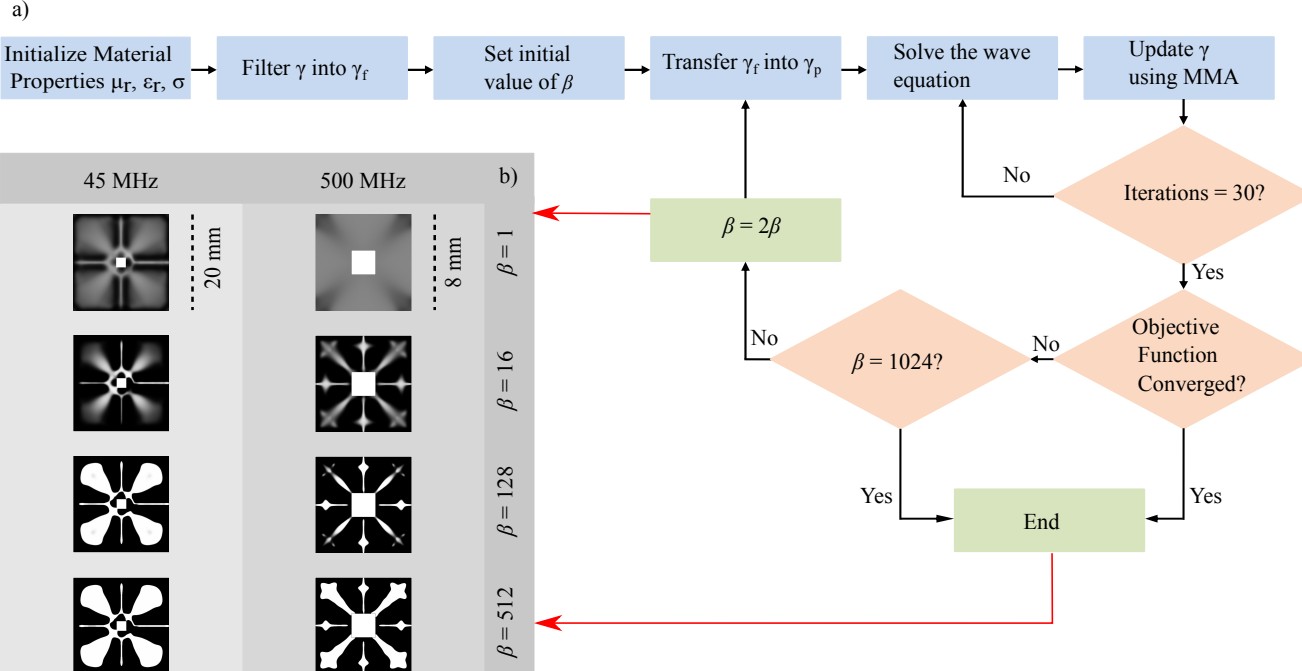

**Figure 3.** (a) Flow chart representing the workflow followed to obtain the material distribution. (b) Intermediate results obtained during the optimization process for different values of frequencies, and projection slope ($\beta$). The material distribution was plotted on a gray scale such that 0 (white) represents air, and 1 (black) represents $Cu$. Final optimization result was obtained at $\beta = 512$, for the designed frequencies.

The computational domains were meshed with linear elements to exploit the efficiency of the linear discretization of the magnetic vector potential. This reduced the total computational cost of the simulations. The simulations were formulated such that they follow an iterative procedure as represented by the flow chart in Figure 3(a). The steps followed sequentially were:

(i) The material properties were initialized in different domains. $\mu_r$, and $\epsilon_r$ were set to unity for the free space and the conductive material. In $\Omega_T$, where the sample would be placed, $\epsilon_r$ was set to 80 and $\mu_r$ to 1, to imitate the electromagnetic behaviour of the water. $\sigma$ was set to zero in all of the domains except in $\Omega_D$, where it was defined as a function of $\gamma_p$, described by Equation 13.





(ii) The initial value of the design variable value was set to be 0.5. It was filtered using Equation 10, and 11. The filtered design variable was then transformed to a projected design variable using a hyperbolic function described in Equation 12. The projection point $\xi$ was fixed at 0.5 and the projection slope $\beta$ initialized at 1, was doubled from its previous value after every 30 iterations.

(iii) The wave equation was solved using the initial value of the design variable after which it was updated using the method of moving asymptotes (*MMA*) (Svanberg (1987)).

(iv) The tolerance for the objective function error was set to be a very small number to allow the conditions of Equation 16 to be met before premature termination of the simulation. If the constraints defined by Equations 16 and 17 were not satisfied till $\beta$ reaches 1024, the computation terminated automatically.

The optimization was performed for two frequencies, *i.e.*, at 45 MHz, and 500 MHz. The results and more detailed setup for these cases are discussed in the following sub-sections.

## 3.1    Optimization of Magnetic Lens at 45 MHz

In an *MR* experiment, the net magnetization of the sample, which is aligned along the $\boldsymbol{B_0}$ field (without loss of generality, the $z$-axis) is flipped by an orthogonal $\boldsymbol{B_1}$ field (without loss of generality, the $x$-axis). Thus, the *MR* coils are designed to produce

a unidirectional $\boldsymbol{B_1}$ field orthogonal to the $\boldsymbol{B_0}$ field. Taking advantage of this condition, we reduced the computational domain to a *2D* geometry. A background magnetic field oscillating at a frequency of 45 MHz along an out of plane vector was defined to imitate the behaviour of the radio frequency magnetic field generated by an *MR* coil.

The background magnetic vector potential becomes $\boldsymbol{A} = (A(z)\exp(i\omega t), 0, 0)$, where the magnitude of A(z) was set to $10^{-3}$ Wbm$^{-1}$, which corresponds to a $\boldsymbol{B_1}$ of $10^{-3}$ T.

The dimension of the entire computational domain was $24 \times 24$ mm$^2$. It was limited by the size of the excitation coil which could fit inside the bore of a 1 T preclinical MRI machine (Bruker ICON). The sample was positioned in a $2 \times 2$ mm$^2$ area in $\Omega_T$, surrounded by $\Omega_D$ where the material interpolation takes place. $\Omega_D$ was enclosed by a 2 mm thick $\Omega_I$.

Figure 3(b) shows the evolution of the topology optimization for intermediate values of $\beta$. The material distribution obtained was plotted on an inverse gray scale $\in [0,1]$ where black corresponding to 1 represents the conductive material and white

corresponding to 0 represents the free space. The *OL* design obtained at the final step, shown in figure 3(c), was used for the subsequent post processing step.

## 3.2    Optimization of Magnetic Lens at 500 MHz

A magnetic lens was also designed for an 11.7 T magnet (Bruker ADVANCE (III) spectrometer), which corresponds to a Larmor frequency of 500 MHz for $^1H$.

The background field was set as defined before in Section 3.1. The computational domain in this case was restricted by the size of a commercially available 10 mm saddle coil, which was used for the verification experiments. The entire computational





domain was $12 \times 12 \text{ mm}^2$. The $\Omega_D$ was truncated by a 2 mm thick $\Omega_I$ to reduce it to a dimension of $8 \times 8 \text{ mm}^2$. The sample was placed in a $2 \times 2 \text{ mm}^2$ area in $\Omega_T$.

Figure 3(b) shows the intermediate results of the topology optimization at every $30^{\text{th}}$ iteration. The material distribution
obtained as shown in Figure 3(c) was plotted on a reverse gray scale as defined in Section 3.1.

## 4   Post Processing

After the designs of *OL*s were obtained, they were characterised and compared with a wired *LL* similar to that discussed by Spengler et al. (2017) and Jouda et al. (2017).

To characterise the magnetic field distribution, and its enhancement, a second simulation environment was set up, where the
background field was replaced by the magnetic field produced by a realistic coil geometry. The boundaries of the *OL*s were truncated by an *IBC*, and electromagnetic proprieties of *Cu* were assigned to it.

An *OL* is designed to enhance a uni-directional magnetic field. it focuses the magnetic field for coils exhibiting this property (Supplementary Figure 2), but for characterization, only a solenoidal coil type was used since the *OL* properties would be similar with the other coil. Figure 4(c), and (d) shows the amplification profile of the magnetic field by the *OL*s in a solenoidal
coil type arrangement. The coils had an outer radius of 23 mm (for 45 MHz), and 6 mm (for 500 MHz) respectively. The two conductive rings were separated by a distance of 3.2 mm; based on the thickness of the *PCB* used for the verification measurements. Figure 4 (b), and (e) represents the current induced in the *OL*s. In Figure 4 (b), the design obtained for the *OL* was asymmetric. At lower frequencies the strength of inductive coupling is weak, therefore the current flow produced by the *Cu* distribution results in the desired magnification while the protrusions maintains the uniformity of the field. As the magnitude
of the electric, and magnetic fields scale linearly with the frequency, at higher frequencies, stronger inductive coupling ensures the enhancement. This forces the algorithm to produce a more symmetric *Cu* distribution to reduce the infeasibilities of the control equations.

To compare the *OL*s with the *LL*s, the geometry similar to as shown in the inset of Figure 4 were used. The total magnetic flux in the target domain without any lens, and with the lens was calculated using the equation

$$B_{total} = \frac{\int_{\Omega_T} \boldsymbol{B}_1 d\Omega_T}{\int_{\Omega_T} 1 d\Omega_T}, \tag{19}$$

For the 45 MHz arrangement, without any lens it was 47.94 µT, and after positioning the *OL* in the coil, the total magnetic flux calculated was 105.31 µT, which resulted in a field magnification of 2.2.

The *OL* was then replaced with a *LL*. The *LL* had a outer diameter of 19 mm, with outer to inner diameter ratio of 5.59. The total magnetic flux calculated for this arrangement was 139.55 µT, which resulted in the field magnification of 2.9.
The *LL* was found to achieve better enhancement of the field compared to the *OL*. However, the field distribution for the *OL* was more uniform. The field uniformity was calculated as the deviation from the $\boldsymbol{B}_1$ at the centre of the lens in a test region $\Omega_T$

$$B_{\text{deviation}}|_{\Omega_T} = \left( \frac{B_{\text{centre}} - B_1}{B_{\text{centre}}} \right) \times 100\% \text{ in } \Omega_T. \tag{20}$$



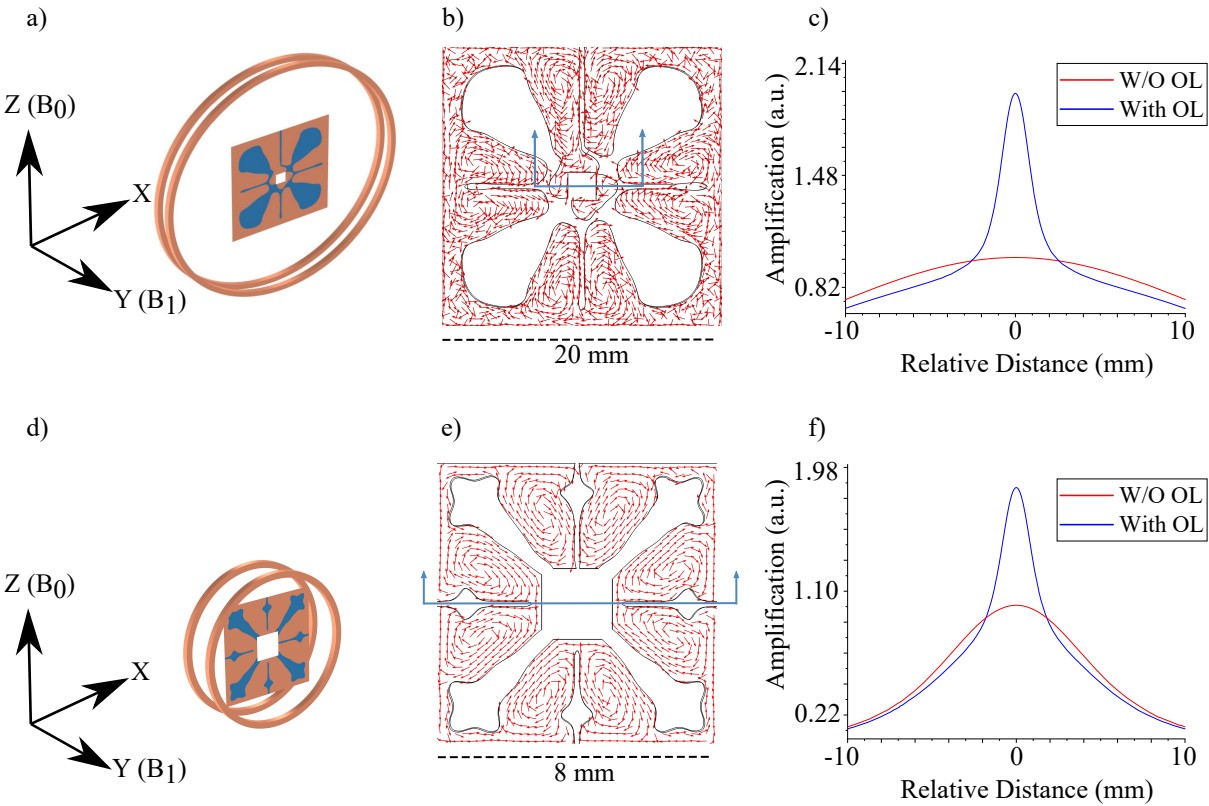

**Figure 4.** (a), and (d), setup used for characterising the *OL* (a) 45 MHz, and (d) 500 MHz. The *OL*s were placed at the centre of two conductive rings such that the magnetic field ($B_1$) produced by the coil is normal to its plane. $B_0$ represents the direction of the static magnetic field to visualize the orientation of the arrangement when placed in the *MR* machine. (b), and (e) show the currents induced in the *OL*. (b) From the current flow it can be interpreted how the magnetic flux would be concentrated at the $\Omega_T$, while the current flowing on copper protrusions ensures the uniformity of the magnetic field. (e) At 500 MHz due to stronger inductive coupling, the amplification produced is high, therefore the optimization results in a symmetric material distribution in order to maintain the uniformity while at the same time ensuring a desired current flow for the field amplification. (c), and (f); amplification produced by the *OL*s, plotted along the direction of $B_1$. The distance represents relative values from the central position of the *OL*s.





For the *OL*, it was 33.72 %, and for the *LL* it was 39.93%. Similarly, the deviation along the central lines were calculated as

$B_{\text{deviation}}|_{L_z}$ and $B_{\text{deviation}}|_{L_x}$. The maximum deviation for the *OL* was calculated to be 20.8% along the *z*-axis, and for the *LL* it was 23.35% along the *x*-axis (see Table 2).

By increasing the frequency of operation for this particular arrangement to 500 MHz, the LL produced a magnification of 3 which is slightly higher than at 45 MHz, and the *OL* produced a magnification of 3.9. As can be seen from the design due to the asymmetric material distribution, with the *OL* the field distribution of the magnetic field was less central. If the region of

interest is reduced such that the variation lies below 10 %, the *OL* at higher frequencies can still be used, therefore depending on the application, one can also use the magnetic lens designed for 45 MHz at 500 MHz to get a higher amplification if maintaining uniformity is not a concern or a smaller sample volume can be used. In order to get a uniform field distribution; following the same protocol for the optimization as described in Sub-section 3.1, and design the magnetic lens at the same dimensional limits for 500 MHz, the eccentricity issue of the previous design was fixed while the magnification obtained was

2. The *OL* obtained for such an arrangement is shown in Figure (Supplementary Figure 1).

Since the magnification of the magnetic field produced by the *LL* depends directly on the ratio of outer to inner diameter. For 1.05 T (Bruker ICON) measurements, the ratio of the coil size to the sample size was large enough to have a higher amplification by the *LL*. If we reduce the size of the coil by keeping the sample dimensions same, as was the case for 11.7 T (Bruker ADVANCE (III)) measurements, where a commercially available Bruker's 10 mm saddle coil was used; this leads to a

reduction in amplification produced by the *LL*. The *OL* designed in Sub-section 3.2, was able to produce a better amplification while maintaining the uniformity.

To verify this, the *OL*, and the *LL* were analysed in a solenoidal coil arrangement. The outer radius of the coil was 6 mm. The two rings were separated by a distance of 3.2 mm. The coils were excited with an AC current oscillating at a frequency of 500 MHz.

The total magnetic flux in the volume of the sample calculated using Equation 20 without any lens was 179.42 µT, and after placing the *OL*, the total flux calculated was 365.65 µT which resulted in amplification factor of 2.04.

By replacing the *OL* with a *LL*, whose outer diameter was 7.6 mm, and outer to inner diameter ratio 2.24; the total magnetic flux was 322 µT which resulted in total amplification factor of 1.79.

Therefore, with the reduction in the outer to inner diameter ratio the total magnification produced by the *LL* is also reduced.

However, the *OL* was still able to maintain the defined amplification. The maximum variation along the central lines calculated from Equation 20, for the *LL* was 19% along the *x*-axis, and for the *OL* was 17% along the *z*-axis. However, total magnetic field variation for both the types were comparable at 33% (see Table 2).

To summarize the discussion, the *LL* was found to have a higher magnification but the field distribution was less uniform. When the ratio of outer to inner diameter for the *LL* is reduced it produces a lower magnification compared to the optimized

magnetic lens albeit, the field uniformity of these devices were similar. Table 2 summarizes the comparison result and Figure 5 shows the amplification profiles for different lenses.



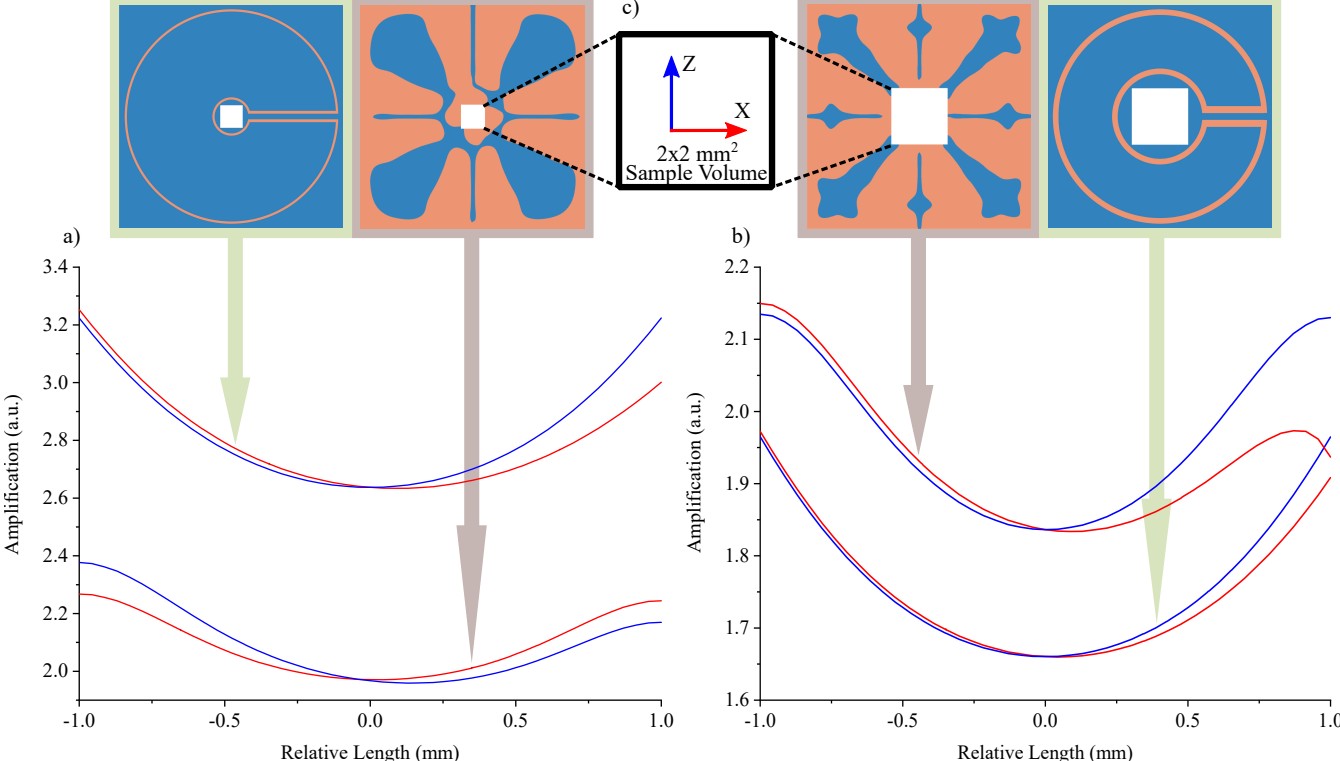

**Figure 5.** Amplification profile comparison of the *OL* with the *LL* in the *x*-*z* plane in $\Omega_T$. (a) The frequency of operation was 45 MHz. The *LL*'s, outer diameter was 19 mm, and the ratio of outer to inner diameter for the *LL* was 5.59. (b) The frequency of operation was 500 MHz. The *LL*'s, outer diameter was 7.6 mm, and the ratio of outer to inner diameter for the *LL* was 2.24. c) The sample region in the centre of the Lenz lenses is always a $2 \times 2$ mm square.

**Table 2.** Comparison summary between *OL*, and *LL*. For the *LL*, the values in brackets represent the ratio of outer to inner diameter.

| Lens | Frequency | Amplification | Variation in $\Omega_T$ | Variation along $L_z$ & $L_x$ |
|------|-----------|---------------|--------------------------|-------------------------------|
| LL(5.59) | 45 MHz | 2.9 | 39.93% | 23.35% |
| Optimized Lens | 45 MHz | 2.2 | 33.72% | 20.8% |
| LL(2.24) | 500 MHz | 1.79 | 33.6% | 19% |
| Optimized Lens | 500 MHz | 2 | 33.4% | 17% |

From the above discussion a question arises; why not set a reference field to achieve an amplification of 5 times rather than a mere factor of 2? The reason a reference field was not set higher is because this leads to the material not being properly distributed, and we get an undefined conductivity values *i.e.*, at gray scale values besides 0 or 1.





## 5   Fabrication and Experimental Verification

After processing the designs from the simulations, they were fabricated, and verified with *NMR* experiments using distilled water as test sample.

### 5.1   Fabrication

The mask for the designs were printed on a butter-paper using a HP Laserjet Enterprise P3015 dn printer.

Using the mask for UV lithographic patterning, the designs were copied onto a positive photosensitised copper board with FR4 laminate with a *PCB* thickness of 1.6 mm, and *Cu* thickness of 35 µm, obtained from an external supplier (C.I.F, France).

The board was etched in a sodium persulfate solution ($Na_2S_2O_8$). The etching solution was prepared such that for every 1 L of DI water, 1 g of $Na_2S_2O_8$ was dissolved in it. The etching solution was placed was in a bubble etch tank (PA104 (Mega Electronics), with etching time varying from 25-60 min depending on the age of the etching solution.

For NMR characterisation, a 0.5 µL sample was prepared in a capillary with an inner diameter of 0.8 mm. The sample occupied a length of 1 mm of the tube.

### 5.2   NMR Experimental Protocol

The low frequency magnetic resonance measurements were acquired by placing the *OL* in a solenoidal coil, whereas for the high frequency measurements a saddle coil was used, as shown in Figure 6(a), and (d).

Both *OL*'s featured self resonance frequencies in the GHz range, as shown in Supplementary Figure 3, therefore combining them with an inductively coupled, and tuned and matched coil, did not effect the overall resonance frequency, nor was the quality factor significantly degraded, see Supplementary Figure 4. The matching conditions were effected, and could be corrected by the probe coil's variable capacitor.

For the *NMR* experiments, all acquisitions were done in single shot without any averaging. The coils were positioned at the iso-centre of the $B_0$ field. The measurements were initialized using the coil without any lens to adjust the power, and the acquisition time. After initialization, a nutation spectrum was acquired to determine the 90° flip angle.

Next, the *OL* was introduced in the coil. The shimming profile had to be re-adjusted, in order to obtain a similar spectral line-width for both experiments. With the same volume of the sample, acquisition time, and power to the coil, a second nutation spectrum experiment was acquired to determine the change in 90° flip angle, and to characterise the $B_1$ uniformity of the *OL*.

The relative intensities of the two arrangements were determined from the areas under the spectrum for a 90° flip angle to show signal enhancement. The noise values were calculated as the deviation of the signal at the baseline of the spectrum, taken in a peak-free region. The *SNR* was calculated as the ratio of the area under the peak signal divided by the noise. Table 3 summarizes the relative *SNR* values, and $B_1$ field enhancement, calculated by Equation 14.

The measurements at 45 MHz proved difficult, mainly due to the large magnetic field drift experienced for the ICON system, exacerbated by the lack of a lock channel on the device. The strategy was to acquire nutation spectra as quickly as possible,





**Figure 6.** (a), and (d) the coil type and *OL* arrangement used for the NMR measurement at 45 MHz. The axis represents the orientation of the device in the NMR measurement apparatus. (b), and (e) fabricated *OL*s on the *PCB*s. (c), and (d) Measured nutation spectra with, and without the *OL*s

i.e., at rather large step size, to estimate the $90°$ flip angle. Nevertheless, through experimental verification we were able to asymptotically determine the magnetic field amplification, and hence the improvement of *SNR* using an *OL* geometry.





**Table 3.** Values calculated from the nutation spectra of water at Larmor frequency of 45 MHz and 500 MHz respectively. The values represent the ratio of the *SNR*, $\boldsymbol{B}_1$ enhancement, and the pulse duration to produce 90° flip angle. The compariso was made with, and without the *OL*.

| Frequency in MHz | Relative SNR | $B_1$ enhancement | 90° pulse duration in µs with *OL* (w/o OL) |
|---|---|---|---|
| **45** | 1.56 | 1.66 | 33.15 (54.9) |
| **500** | 1.19 | 1.3 | 12.5 (16.23) |

## 6   Conclusions

It is hardly a surprise that the quest for more signal-to-noise from an existing NMR detector arrangement is matter of numerical optimization. Topology optimization offers a feasible pathway with which to reach optimal designs that goes beyond mere intuition, and we could show, using a commercial finite element tool, that it is possible to find practical Lenz lens arrangements that, when implemented, achieve their set goals. The found topologies form a compromise between signal enhancement and field uniformity. Of course it would have been possible to extend the Pareto front to include additional goals, such as maintaining a good susceptibility shift profile. Our experience is, however, that this leads to difficulties, mainly because the optimization problem becomes over-constrained, and hence no longer evolves towards useful design modifications.

We only discussed the use of optimization to enhance the magnetic field of an *MR* coil, but of course the methodology could be further extended to designing a self resonant structure, in order to avoid the tedious task of matching and tuning. In this regard, the designs found reconfirmed one useful aspect of Lenz lens arrangements, namely, that they do not modify the tuning of the outer driver coil, merely its matching condition (the depth of the absorption dip in the $S_{11}$ curve), and suitable reflection conditions are easily found.

An important aspect is the ability to achieve manufacturable designs. For the case where a design is essentially a two-dimensional metal patch on a dielectric sheet, printed circuit boards are an inexpensive route towards implementation, easy to manufacture, and lead to satisfying result for arbitrary embedded topologies. However, for the case of 3D topologies, the situation is quite different and not all found geometries will be manufacturable. For example, to minimize eddy current losses, designs tend to evolve towards tiny disconnected islands of metal, arranged in a dielectric background, which would require very advanced 3D printing to achieve. We did not pursue such designs in this contribution, but the message should be clear. Optimization must include manufacturing constraints, in order to achieve feasible designs.

Beyond manufacturability, a design must be practicable in use, which further limits the design freedom, because the sample must be provided with a convenient way in and way out of the sensitive volume of the detector. We found no problems with 2D designs, but 3D designs posed a challenge, resulting in designs that left too little space.

The general conclusion is that all aspects of a design must be mathematically expressable as a goal function in order to be considered, but that, as more terms are added to the optimization goal, the numerical convergence process slows down, eventually reaching a standstill.

Because computational electromagnetics is scale invariant, the topology optimization methodology is applicable to resonator
arrangements beyond the range of typical nuclear magnetic resonance frequencies, applications such as the design of magnetic or electric resonators used in EPR, the optimization of individual wavepath components including capacitors and striplines, and even for wireless energy transfer.

*Author contributions.* Concept initialisation: SW, DM, and JGK. Topology Optimization and Simulation setup: SW, YD, and JGK. Fabrication: SW. Initial measurements: SW and ON. Measuerements: MJ, and SW. Funding request and supervision: JGK. Writing, review, and
editing: SW, MJ, YD, DM, and JGK.

*Competing interests.* The authors declare no competing interests

*Acknowledgements.* The authors would like to thank the Karlsruhe Institute of Technology for its continued support and also for providing a safe working environment during the *COVID-19* pandemic. SW and JGK acknowledges partial financial support by the 'Virtual Materials
Design' (VIRTMAT) initiative at KIT. We sincerely acknowledge the Deutsche Forschungsgemeinschaft for partially supporting this research under contract KO 1883-20 Metacoils. JGK and MJ acknowledge the DFG for partial funding (Contract KO 1883/29-1). DM and JGK acknowledge additional support from the EU2020 FET grant (TiSuMR, 737043). YD acknowledges support from a Humboldt Research Fellowship for Experienced Researchers (Humboldt-ID: 1197305). ON acknowledges support from the German Academic Exchange Service (DAAD) under the German Egyptian Research Long-Term Scholarship Program (GERLS). We would also like to use this opportunity to
express our gratitude towards the *BW-Uni* Cluster for allowing to use their resources, and acknowledge Dr. Ralf Ahrens for providing access to the computer used for the computation.




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
