# Peer review of "Topologically Optimized Magnetic Lens for MR Applications"

_Magnetic Resonance, 2020_

## Referee Comment (RC1) · Anonymous Referee #1 · 31 Aug 2020

The manuscript describes the design and implementation of a technique to increase the amount of RF power delivered to the sample region in magnetic resonance experiments. The B1 coils in MR experiment do a two fold job: a) Creating a homogeneous magnetic field pulse along a direction perpendicular to the main magnetic field direction (taken to be the z -direction in general), b) Detecting the flux induced by the larmor precession of sample magnetisation, the main signal in a MR experiment. The signal to noise measured in any MR experiment thus critically depends on the efficiency of this coil to do both its jobs. In this manuscript a magnetic lens design is computationally optimised such that it focuses more of the rf power from the B1 coil into the sample volume. The optimal design is also fabricated and tested to check for power enhancement at the sample.

[Figure]

Improving signal to noise(S/N) and resolution are a constant objective of new techniques introduced in MR. This manuscript provides a way to augment the hardware in exciting MR set-ups to increase S/N by about a factor of two. The technique derives its working principle from the Lenz lens used to enhance the effective filling factor of sample in receiver coils in MR. The manuscript shows a way to computational optimise the lens design to get the most out of the system. Though the enhancement achieved in the end is not very different from the standard Lenz lens design, given the constrains of construction, it demonstrates the possibility to use such a software optimisation for building coils with specific deliverables. They use commercial finite element analysis software for the task. The method can be modified to design coils with other specification such as broad tunability or homogeneity over a specified region of interest etc. Thus the manuscript demonstrates the possibility to use this technique for hardware improvements in MR receiver coil design.

The article starts by introducing the need for S/N improvements in MR and describes some of the hardware techniques that have been used in literature for enhancing sample filling factor and rf power at the sample. **It would be nice to also talk about typical homogeneity in rf fields of the various coils current in use.** The Lenz lens(LL) is introduced and the need to optimising the LL lens to achieve specific goals is mentioned. The background of topology optimisation for inverse material design is also introduced with relevant references.

The main aspects of the new design concern the electromagnetic wave propagation through medium and the electrodynamics involved in magnetic induction and wave propagation through interface. The relevant electromagnetic equation: Maxwell equations and its modifications for the problem are properly introduced. The boundary conditions under which the equations are solved for also explained. **The parameter**

**called penetration factor** $p$ **needs to explained.** They provide a comprehensive list of constrains and conditions under which they solve for the optimisation. **It would be nice to know if these conditions are sufficient in themselves, may be with some representative examples from literature.**

The optimisation protocol basically creates a region of metal (Cu) and air and finds the profile of Cu that would give rise to the required field strength and uniformity at the sample region. A design variable $\gamma$ is used to define free space or Cu [0 or 1]. They use a hyperbolic tangent function, defined by a parameter $\beta$, to define the material contrast between the Cu and air regions such that construction of the coil would be practical without region defined in multiple grayscale. The numerical algorithm is set up in a commercial software - COMSOL. The various steps for calculating the optimal lens (OL) by iterating over $\gamma$ and $\beta$ is explained starting from a initial shape with uniform $\gamma$ and linearly boundaries ($\beta$ =1). **The $\beta$ values are course grained as each increment of $\beta$ doubles its value. Would a 'slower' increment give better optimisation or the boundary shape change is still sufficiently slow with this step size?**

The details of the procedure for obtained a OL lens at 45MHz and 500 MHz are explained. Post processing simulations to check the field enhancements given the OL are also performed and the results are explained well. The enhancement obtained are compared to standard receiver colis in their MR instruments for OL and LL and the results are well tabulated in Table 2. The inhomogeneity in the magnetic field are also simulated and compared. They also justify the fact that the amplification achieved was not very different between OL and LL due to practical constrains.

They go ahead and fabricate the LL and perform NMR nutation experiments to demonstrate the rf field enhancement and the results are well tabulated in Table 3.

The paper successfully demonstrates that commercial finite element software can be used to find practical, optimal LL lens with set goals even if the OL does not show much difference from LL.

The theory, the numerical algorithm, simulation and the NMR experiments are all explained well and figures and tables are well presented. The language of the text though requires to be improved. There are many grammatical and typographical errors in the article that need to be taken care of. I will attempt to list a few below.

**Technical Corrections**

1. Abstract: Line 4 current *"...designs that delivers..."* change to "*designs that deliver..."*

2. Abstract: Line 5 current *"...an RF circuit..."* change to *"a RF circuit..."*

3. Introduction: Line 13 current *"... and also leads to a reduction in ...."* change to *"... and also a reduction in ...."*

4. Introduction: Line 43 current *"... by Jauga et. al that by tuning and matching ...of operation, this improved...."* change to *"... by Jauga et. al that tuning and matching .......of operation, improved..."*

5. Methodology: Line 103 current *"....blue region in te centrerepresents ...."* change to *"....blue region in tne center represents ...."*

6. Methodology: Line 124 current *".. any values..."* change to *" any value"*

7. Post processing: Line 212 current *"...magnetic field. it focuses..."* change to *"........magnetic field. It focuses..........."*

8. Post processing: Line 219 current *"...while the protrusions maintains the uniformity of the field..."* change to *".......while the protrusions maintain the uniformity of the field.........."*

9. Post processing: Line 223 current *"... the geometry similar to as shown in ..."* change to *"....... ageometry similar to that shown in .........."*

10. Fabrication: Line 278 current *"...... solution was placed was in a bubble ....."* change to *"...... solution was placed in a bubble ........"*

11. Conclusions Line 307 current *"......The found topologies form a ....."* change to *"......The topologies found by optimization form a ..........."*

---

## Author Comment (AC1) · 7 Sep 2020

[journal abbreviation, manuscript]copernicus

[]Wadhwa et al.

Jan G. Korvink (jan.korvink@kit.edu)
**Interactive Comment on "*Topologically Optimized Magnetic Lens for MR Applications*" by Wadhwa et al.**

September 7, 2020

Authors: We would like to thank the reviewer for the time and patience for the detailed review of our manuscript. We hope that our comments and the revised manuscript will meet the reviewer's expectations.

Reviewer: The manuscript describes the design and implementation of a technique to increase the amount of RF power delivered to the sample region in magnetic resonance experiments. The B1 coils in MR experiment do a two fold job: a) Creating a homogeneous magnetic field pulse along a direction perpendicular to the main magnetic field direction (taken to be the z-direction in general), b) Detecting the flux induced by the larmor precession of sample magnetisation, the main signal in a MR experiment. The signal to noise measured in any MR experiment thus critically depends on the efficiency of this coil to do both its jobs. In this manuscript a magnetic lens design is computationally optimised such that it focuses more of the rf power from the B1 coil into the sample volume. The optimal design is also fabricated and tested to check for power enhancement at the sample.

Improving signal to noise(S/N) and resolution are a constant objective of new techniques introduced in MR. This manuscript provides a way to augment the hardware in exciting MR set-ups to increase S/N by about a factor of two. The technique derives its working principle from the Lenz lens used to enhance the effective filling factor of sample in receiver coils in MR. The manuscript shows a way to computational optimise the lens design to get the most out of the system. Though the enhancement achieved in the end is not very different from the standard Lenz lens design, given the constrains of construction, it demonstrates the possibility to use such a software optimisation for building coils with specific deliverables. They use commercial finite element analysis software for the task. The method can be modified to design coils with other specification such as broad tunability or homogeneity over a specified region of interest etc. Thus the manuscript demonstrates the possibility to use this technique for hardware improvements in MR receiver coil design.

The article starts by introducing the need for S/N improvements in MR and describes some of the hardware techniques that have been used in literature for enhancing sample filling factor and rf power at the sample. **It would be nice to also talk about typical homogeneity in rf fields of the various coils current in use**.

Authors: We have added the information about the magnetic field homogeneity of the coils that were used for the post processing of the Optimized Lens in lines 242 and 270 of the revised manuscript.

Reviewer: The Lenz lens(LL) is introduced and the need to optimising the LL lens to achieve specific goals is mentioned. The background of topology optimisation for inverse material design is also introduced with relevant references.

The main aspects of the new design concern the electromagnetic wave propagation through medium and the electrodynamics involved in magnetic induction and wave

propagation through interface. The relevant electromagnetic equation: Maxwell equations and its modifications for the problem are properly introduced. The boundary conditions under which the equations are solved for also explained. **The parameter called penetration factor p needs to explained.**

Authors: To answer the reviewer's request we assume that, the penalization factor *p* was referred to as the penetration factor in Equation 13. If our assumption is correct, the penalization factor was used in the conductivity function to assign a conductivity value for different values of $\gamma_p$, as shown in Figure 2 (b) of the manuscript. By changing the value of *p*, the range of conductivity values corresponding to the range of $\gamma_p$ can be altered, as shown in Figure 1 of this document, where the shaded regions correspond to $\gamma_p \in$ [4.5,5.5]. For lower values of *p* the design obtained will contain grayscale conductivity values that will not have any physical meaning. To reduce the grayscale it was necessary to increase *p*. However, for higher values of *p*, the material would be assigned as *Cu* even for values of $\gamma_p$ close to zero (i.e. $\gamma_p \approx 0$ ). The maximum value up to which *p* could be increased was based on trial and error method, and was found to be optimal for *p* = 3. We have added this information in the revised manuscript between lines 134-139 and line 140. Figure 1 is added to the Supplementary of the manuscript.

Reviewer: They provide a comprehensive list of constrains and conditions under which they solve for the optimisation. **It would be nice to know if these conditions are sufficient in themselves, may be with some representative examples from literature**.

Authors: The control equations were defined such that the *Cu* distribution obtained tries to satisfy two conditions, i.e. the field enhancement, and its homogeneity in the sample region. During the numerical experiments, we did not find other conditions that might have helped to improve these goals, without over-constraining the optimization computation. The results obtained are an indication that the control equations were

sufficient to produce the desired results, though the uniformity was not as good as the coil itself, as these two conditions were in conflict with each other as mentioned in line 52 of the revised manuscript. When this reply was written, even after a thorough investigation we could not find any supporting literature where topology optimization was used for the scenario presented in the manuscript i.e. optimization of RF devices for field enhancement while maintaining homogeneity.

Reviewer: The optimisation protocol basically creates a region of metal (Cu) and air and finds the profile of Cu that would give rise to the required field strength and uniformity at the sample region. A design variable $\gamma$ is used to define free space or Cu [0 or 1]. They use a hyperbolic tangent function, defined by a parameter $\beta$, to define the material contrast between the Cu and air regions such that construction of the coil would be practical without region defined in multiple grayscale. The numerical algorithm is set up in a commercial software - COMSOL. The various steps for calculating the optimal lens (OL) by iterating over $\gamma$ and $\beta$ is explained starting from a initial shape with uniform $\gamma$ and linearly boundaries ($\beta$=1). **The $\beta$ values are course grained as each increment of $\beta$ doubles its value. Would a 'slower' increment give better optimisation or the boundary shape change is still sufficiently slow with this step size?**

Authors: The reviewer's observation is correct in assuming that, with the slower increment of $\beta$ the hyperbolic tangent function's shape change was slow. The $\beta$ value used in Equation 12 of the manuscript was used to transform this equation from a linear (for lower values of $\beta$) to a unit step function (for higher values of $\beta$) as shown in Figure 2(a) of the manuscript. This was done to produce a high contrast material distribution, as $\gamma_p$ is forced to be either air or *Cu* for higher values of $\beta$. The reason for not choosing a smaller step size was that the filtered design variable, $\gamma_f$ would have been projected by a similar function and this would have increased the iteration steps without improving the optimization process (We have added this information between

lines 179-182 of the revised manuscript).

Reviewer: The details of the procedure for obtained a OL lens at 45MHz and 500 MHz are explained. Post processing simulations to check the field enhancements given the OL are also performed and the results are explained well. The enhancement obtained are compared to standard receiver colis in their MR instruments for OL and LL and the results are well tabulated in Table 2. The inhomogeneity in the magnetic field are also simulated and compared. They also justify the fact that the amplification achieved was not very different between OL and LL due to practical constrains.

They go ahead and fabricate the LL and perform NMR nutation experiments to demonstrate the rf field enhancement and the results are well tabulated in Table 3.

The paper successfully demonstrates that commercial finite element software can be used to find practical, optimal LL lens with set goals even if the OL does not show much difference from LL.

The theory, the numerical algorithm, simulation and the NMR experiments are all explained well and figures and tables are well presented. The language of the text though requires to be improved. There are many grammatical and typographical errors in the article that need to be taken care of. I will attempt to list a few below......

Authors: We apologise for these mistakes and have corrected them. With all due respect to the reviewer's suggestions, correction #2 was not made. The choice between "a" or "an" was based on the vowel sound of the word 'RF' as suggested in Merriam-Webster: https://www.merriam-webster.com/words-at-play/is-it-a-or-an, last access: 03 September 2020. All changes made are indicated in blue in the revised manuscript.

**Fig. 1.** Response of the conductivity function for different values of the penalization factor. The shade regions corresponds to the conductivity values for $\gamma_p \in [4.5, 5.5]$. [FIGURE IS SEPARATELY UPLOADED]

---

## Referee Comment (RC2) · Anonymous Referee #2 · 8 Sep 2020

Sensitivity is the bane of nuclear magnetic resonance spectroscopy and any improvement in the signal to noise ratio (SNR) is welcome. The paper under review is in that vein; it discusses the design, by computer optimization, of a "distributed metal track" (in the words of the authors) to serve as a Lenz lens. A Lenz lens, called so because it uses Lenz's law of induction and like a lens focusses the magnetic field by a larger coil into a smaller region, when placed between the RF circuit and the sample improves the SNR in MRI and NMR by engineering the distribution of the magnetic field intensity in a region of interest. The paper demonstrates the optimal topology of a Lenz lens, which is then fabricated and its use validated in MR applications at 45 MHz and 500 MHz.

The approach adopted is topology optimization, which answers the question "how to

place material within a prescribed design region for optimal performance?" Topology optimization is achieved by the use of an adjustable, spatially varying material property. The authors selected the conductivity of the medium to be a function of the spatial coordinates, which they allowed to vary between that of free space, and Cu, a range of $10^e7$. The uniformity of the RF field, which determines the flip angle of the pulse, was the control equation which was minimized subject to a number of constraints. Following the design of two optimal Lenz lenses (one operating at 45 MHz and another at 500 MHz) using a commercial finite element software package, in a second simulation stage called post-processing, the magnetic field distribution in the lenses were characterized by replacing the background field with the magnetic field produced by a realistic coil geometry. Finally, the lenses were fabricated and their performance verified with nutation experiments on a water sample at 45 MHz in a 1 T pre-clinical MRI machine and at 500 MHz in a high resolution NMR spectrometer.

They conclude that topology optimization, using a commercial finite element tool, offers a feasible way to find practical Lenz lens arrangements which can be easily fabricated. They were able to find lenses with a $B_1$ enhancement of about 1.5, a marginal increase in SNR (1.2 and 1.6), a modest decrease in the $\pi/2$ pulse duration, and that the topologies are a compromise between signal enhancement and field uniformity.

The improvements demonstrated are modest and restricted to 2D, but the approach shows promise. The mechanics of the paper (the English, punctuations, capitalization, etc.) is, however, in need of some serious repair; errors of this nature are too numerous to even list. The authors are also encouraged to consider some other questions/comments given below.

1. In the introduction, the two sentences on filling factor appear to be in disagreement.

2. Chemical elements, such as Cu, are not italicized.

[Figure]

3. The acronym OL is not defined.

4. What are the limitations of restriction to 2D geometry? The sample volumes are much smaller than is practically used so any gain in SNR is compromised because of smaller sample volume. Besides, when you are off-resonance, the trajectory of magnetization is not 2D anyway. Some discussion on this would be welcome. Would having two 2D lenses at the two ends of a solenoid be useful? Was it considered?

5. In Figure 5, what does it mean to have relative length in mm? Would it not be dimensionless?

6. It is BRUKER AVANCE not ADVANCE.

7. Figures 6 c) and 6 f) are not nutation spectra; their Fourier transform is. Why is there an asymmetry (below and above the maximum) in Figure 6 c) with OL?

8. A number of points in the conclusions (3D geometry, Eddy currents, tuning and matching) merit some elaboration in the manuscript. The statement "It is hardly a surprise that the quest for more signal-to-noise from an existing NMR detector arrangement is matter of numerical optimization." is questionable. Pardon my ignorance, but I do not know what a "Pareto front" is.
* * *
Interactive comment on Magn. Reson. Discuss., https://doi.org/10.5194/mr-2020-17, 2020.

---

## Author Comment (AC2) · 17 Sep 2020

[journal abbreviation, manuscript]copernicus

[]Wadhwa et al.

Jan G. Korvink (jan.korvink@kit.edu)
**Interactive Comment on "*Topologically Optimized Magnetic Lens for MR Applications*" by Wadhwa et al.**

September 17, 2020

Authors: We would like to thank the reviewer for taking out time to review our manuscript. We welcome the comments made, and hope the response to the questions raised will meet the reviewer's expectations.

Reviewer: Sensitivity is the bane of nuclear magnetic resonance spectroscopy and any improvement in the signal to noise ratio (SNR) is welcome. The paper under review is in that vein; it discusses the design, by computer optimization, of a "distributed metal track"(in the words of the authors) to serve as a Lenz lens. A Lenz lens, called so because it uses Lenz's law of induction and like a lens focusses the magnetic field by a larger coil into a smaller region, when placed between the RF circuit and the sample improves the SNR in MRI and NMR by engineering the distribution of the magnetic field intensity in a region of interest. The paper demonstrates the optimal topology of a Lenz lens, which is then fabricated and its use validated in MR applications at 45 MHz and 500MHz.

The approach adopted is topology optimization, which answers the question "how to

place material within a prescribed design region for optimal performance?" Topology optimization is achieved by the use of an adjustable, spatially varying material property. The authors selected the conductivity of the medium to be a function of the spatial co-ordinates, which they allowed to vary between that of free space, and Cu, a range of10e7. The uniformity of the RF field, which determines the flip angle of the pulse, was the control equation which was minimized subject to a number of constraints. Following the design of two optimal Lenz lenses (one operating at 45 MHz and another at500 MHz) using a commercial finite element software package, in a second simulation stage called post-processing, the magnetic field distribution in the lenses were characterized by replacing the background field with the magnetic field produced by a realistic coil geometry. Finally, the lenses were fabricated and their performance verified with nutation experiments on a water sample at 45 MHz in a 1 T pre-clinical MRI machine and at 500 MHz in a high resolution NMR spectrometer.

They conclude that topology optimization, using a commercial finite element tool, offers a feasible way to find practical Lenz lens arrangements which can be easily fabricated. They were able to find lenses with a $B_1$ enhancement of about 1.5, a marginal increase in SNR (1.2 and 1.6), a modest decrease in the $\pi/2$ pulse duration, and that the topologies are a compromise between signal enhancement and field uniformity.

The improvements demonstrated are modest and restricted to 2D, but the approach shows promise. The mechanics of the paper (the English, punctuations, capitalization, etc.) is, however, in need of some serious repair; errors of this nature are too numerous to even list. The authors are also encouraged to consider some other questions/comments given below.

1. Reviewer: In the introduction, the two sentences on filling factor appear to be in disagreement.

    Author: We apologise for this. What we meant by the statement was that, due to the filling factor (geometrical relation between the coil and the sample volume),

the maximum usable sample volume is limited by the size of the effective $B_1$-volume of the coil. But if the sample volume happens to be smaller than the size of the coil, the filling factor can also be improved by reducing the coil's size. We have rephrased the sentences and the changes are indicated between lines 19-25 of the revised manuscript.

2. Reviewer: Chemical elements, such as Cu, are not italicized.

    Author: We thank the reviewer for pointing this out. The chemical elements written in Italic type have been replaced by the suggestion from the Copernicus Publications using LaTeX command `\chem{}`. The changes are indicated in the revised manuscript.

3. Reviewer: The acronym OL is not defined.

    Author: We have now defined the acronym *OL* in the revised manuscript's line 77.

4. Reviewer: What are the limitations of restriction to 2D geometry?

    Author: By reducing the computation to a 2D problem we restricted the material distribution on a plane normal to the $B_1$ direction. This was done because from Maxwell's equations, it is known that the curl of the current gives the direction of the magnetic field; therefore, to get a unidirectional magnetic field, the material interpolation would have been dominant on this plane. One of the limitations with this approach is that the magnetic field amplification was not along the entire cross-sectional length of the sample, as shown in the graphs in Figure 4 of the manuscript. A 3D design evolution could have improved it. Additionally, 3D topology optimization has the potential to further improve the field homogeneity, since the design will have an extra degree of freedom for material interpolation.

    The inverse material design in 3D space comes with its challenges as discussed in the Conclusion of the manuscript. One major issue is to fabricate the design

obtained. Another issue is that the design domain will be restricted to allow for sample placement. Adding additional control equations to overcome these problems over-constrain the optimization problem resulting in a non-converging computation. We still need to work on finding proper conditions, which can provide a useful 3D geometries.

Reviewer: The sample volumes are much smaller than is practically used so any gain in SNR is compromised because of smaller sample volume.

Author: We agree with the reviewer that a smaller sample volume will degrade the *SNR* of the acquired signal. We do not conflict the fact that, in a conventional NMR spectroscopy or imaging, the sample volume used is much higher than the volume used by us for validating the simulation results. Through this paper, we wanted to demonstrate that for special cases for e.g., when performing *MRI* on small living organisms, or for sensitive spectroscopy of small samples, where the coil cannot be placed near the sample, as stated in line 30 of the revised manuscript; it is still possible to enhance the *SNR* of the system by improving the filling factor of the coil. One way to achieve this was by focusing the magnetic field generated by the coil in the sample region. To find the optimum material distribution which could fulfil the requirements *i.e.,* field enhancement (focusing of the magnetic field) and field uniformity, we used topology optimization to obtain the distribution of Cu. Therefore, to compare the performance of the coils with, and without the *OLs*, the sample volume was kept constant to calculate the improvement in the *SNR*.

Reviewer: Besides, when you are off-resonance, the trajectory of magnetization is not 2D anyway. Some discussion on this would be welcome.

Author: To answer the comment made by the reviewer, we have assumed two scenarios:

a)**The coil and Lenz lens arrangement are not tuned to the frequency of operation**
As mentioned by the reviewer "the trajectory of magnetization is not 2D..". The magnetization will have three components in this case. However, only the magnetization component in $I_x$ and $I_y$ direction will contribute to the *NMR* signal, any magnetization along $I_z$ is irrelevant. The *NMR* coils are designed to deliver a unidirectional magnetic field ($B_1$); therefore, by the reciprocity principle of Maxwell's equations, the coil will only detect the magnitude of the magnetization projected along the $B_1$ direction. Similarly, the Lenz lens enhances the magnetic field in the direction of $B_1$; therefore, the Lenz lens will only enhance the signal produced due to the magnetization in the direction of $B_1$. Though, the signal acquired will be weak if the detector/receiver arrangement is not matched and tuned. This would also be the case when a sufficient volume of the sample is used, but the coil is not matched and tuned at the Larmor frequency.

b)**Lenz lenses are used at frequencies; besides, for the one they have been designed for *i.e.,* 45 MHz, and 500 MHz**

This will not have any effect in the field enhancement, or as a matter of fact on the tuning condition of the coil (as mentioned in line 38 of the revised manuscript); though, the field uniformity may be disturbed. The Lenz lenses are broad-band up to their resonance frequency (as mentioned in line 37 of the revised manuscript). The geometry obtained for low-frequency will also enhance the magnetic field if used at higher frequencies. The field enhancement improves with the increasing frequency (due to stronger inductive coupling) but the uniformity degrades due to the asymmetric material distribution (mentioned in line 247 of the revised manuscript). For the high-frequency geometry, the structure obtained is symmetric, and so is the field distribution with the enhancement. If this geometry is used at lower frequencies the field enhancement will be poor compared to the low-frequency geometry. The characterisation of the lenses at different frequencies was not done, since it was assumed that, if required the operator would design the lens at the frequency of operation to get a better performing device. By better, we mean the lens which produces the magnetic field enhancement, whilst maintaining field uniformity.

Reviewer: Would having two 2D lenses at the two ends of a solenoid be useful? Was it considered?

Author: This is an interesting observation by the reviewer. Ideally, a Lenz lens should also amplify the field in the rings of the solenoid, which should improve its performance, but it's not straightforward. This condition was already explored and reported by Spengler et al. (2017). It does improve the homogeneity further, but only if it forms a kind of a Helmholtz coil pair. Nevertheless, the individual *LL* must also be uniform in order to achieve this condition, and hence, that was why we chose the optimization of a single *LL* as a goal.

5. Reviewer: In Figure 5, what does it mean to have relative length in mm? Would it not be dimensionless?

Author: We apologise for this confusion. The graph in Figure 5, represents the amplification profile in the x-direction (red), and z-direction (blue), where the horizontal axis represents the relative distance from the centre point of the *OLs*. We have added this information in the caption of Figure 5.

6. Reviewer: It is BRUKER AVANCE not ADVANCE

Author: We apologise for this typo and have corrected them in the revised manuscript.

7. Reviewer: Figures 6 c) and 6 f) are not nutation spectra; their Fourier transform is.

Author: Agreeing with the reviewer's comment we have modified the caption of Figure 6 to include that the graphs represent the Fourier transform of the nutation spectra.

Reviewer: Why is there an asymmetry (below and above the maximum) in Figure 6 c) with OL.

Author: As stated in line 310 of the revised manuscript, there was a large magnetic field drift experienced for the ICON system, which was aggravated due to the lack of a frequency locked channel. This causes issue for the long measurements, which was the reason for the asymmetrical nutation spectra.

8. Reviewer: A number of points in the conclusions (3D geometry, Eddy currents, tuning and matching) merit some elaboration in the manuscript.

Author: We have elaborated the points mentioned by the reviewer in the revised manuscript.

Reviewer: The statement "It is hardly a surprise that the quest for more signal-to-noise from an existing NMR detector arrangement is matter of numerical optimization." is questionable.

Author: We apologise for the confusion that our statement might have created. What we meant was that in an NMR detector, by adding a passive element (in this case a Lenz lens), if all other conditions *i.e.,* the volume, the current applied to the coil, etc. are kept constant, the numerical optimization can help to find an optimum design for such passive elements to enhance the *SNR*. We have rephrased this sentence in the revised manuscript.

Reviewer: Pardon my ignorance, but I do not know what a "Pareto front" is.

Author: "Pareto efficiency" is a concept named after "Vilfredo Pareto". It is used to define a situation where an objective cannot be improved further without affecting the other objective or objectives.

For the explanation, if we have a bi-objective model, where the objective functions are represented on the two orthogonal axes, the line connecting the current objective values forms a front. Improving one objective, generally worsens the other

*i.e.,* the "Pareto Front" shows the compromise. One can then choose between different values of the objectives to find the best compromise (Jones,Dylan and Tamiz,Mehrdad (2010)). We have added the reference in the revised manuscript

**References**

Jones, Dylan and Tamiz, Mehrdad: Practical Goal Programming, in: International Series In Operations Research and Management Science (volume 141), edited by: Hillier, Frederick S., Springer, New York, Dordrecht, Heidelberg, London, 2010, https://doi.org/10.1007/978-1-4419-5771-9.